# Storage and Utilization of Glycogen by Mouse Liver during Adaptation to Nutritional Changes Are GLP-1 and PASK Dependent

**DOI:** 10.3390/nu13082552

**Published:** 2021-07-26

**Authors:** Ana Pérez-García, Verónica Hurtado-Carneiro, Carmen Herrero-De-Dios, Pilar Dongil, José Enrique García-Mauriño, María Dolores Sánchez, Carmen Sanz, Elvira Álvarez

**Affiliations:** 1Department of Biochemistry and Molecular Biology, Faculty of Medicine, Complutense University of Madrid, 28040 Madrid, Spain; anpere07@ucm.es (A.P.-G.); pilar.dongil@uam.es (P.D.); eao513@ucm.es (E.Á.); 2The Health Research Institute of the Hospital Clínico San Carlos, Ciudad Universitaria, 28040 Madrid, Spain; 3Department of Cell Biology, Faculty of Medicine, Complutense University of Madrid, 28040 Madrid, Spain; cherrerod@salud.madrid.org (C.H.-D.-D.); jegmaurino@med.ucm.es (J.E.G.-M.); mdolosan@ucm.es (M.D.S.); mcsanz@ucm.es (C.S.); 4Department of Physiology, Faculty of Medicine, Complutense University of Madrid, 28040 Madrid, Spain

**Keywords:** exendin-4, glucose, metabolic sensors, diabetes, glucokinase, fasting, feeding

## Abstract

Glucagon-like peptide 1 (GLP-1) and PAS kinase (PASK) control glucose and energy homeostasis according to nutritional status. Thus, both glucose availability and GLP-1 lead to hepatic glycogen synthesis or degradation. We used a murine model to discover whether PASK mediates the effect of exendin-4 (GLP-1 analogue) in the adaptation of hepatic glycogen metabolism to nutritional status. The results indicate that both exendin-4 and fasting block the *Pask* expression, and PASK deficiency disrupts the physiological levels of blood GLP1 and the expression of hepatic GLP1 receptors after fasting. Under a non-fasted state, exendin-4 treatment blocks AKT activation, whereby Glucokinase and Sterol Regulatory Element-Binding Protein-1c *(Srebp1c)* expressions were inhibited. Furthermore, the expression of certain lipogenic genes was impaired, while increasing Glucose Transporter 2 (GLUT2) and Glycogen Synthase (GYS). Moreover, exendin-4 treatment under fasted conditions avoided Glucose 6-Phosphatase *(G6pase)* expression, while maintaining high GYS and its activation state. These results lead to an abnormal glycogen accumulation in the liver under fasting, both in PASK-deficient mice and in exendin-4 treated wild-type mice. In short, exendin-4 and PASK both regulate glucose transport and glycogen storage, and some of the exendin-4 effects could therefore be due to the blocking of the *Pask* expression.

## 1. Introduction

The liver plays a central role in metabolic homeostasis, considering its function in the storage and redistribution of carbohydrates, proteins and lipids. It is an especially vital organ in the adjustment to nutritional changes under both feeding and fasting conditions. The liver is therefore one of the main organs involved in glycogen synthesis and degradation [1]. Thus, during feeding periods when the uptake of glucose occurs, liver stores it as glycogen, helping to control glycemia. By contrast, the liver produces glucose under fasting conditions, first by glycogenolysis and then through hepatic gluconeogenesis, as the main fuel source for other tissues [2,3]. 

Glycogen synthesis and degradation processes are highly regulated by enzymes coordinated with hormonal control that adjust glycogen metabolism to the entire body’s glucose availability and demand [4].

When blood glucose levels rise, cells uptake glucose thanks to the presence in the liver of the glucose transporter GLUT2. Glucose is then immediately phosphorylated by hexokinases, especially glucokinase (GCK), which acts as a glucose sensor, as it has an enzymatic activity with a high Km and is not saturated by the product (glucose 6-phosphate). When intracellular glucose is high, glycogen synthase (GYS) is one of the critical enzymes in glycogen synthesis. It is regulated by phosphorylation by PKA, GSK3 and others, such as PASK, which phosphorylate GYS at Ser-640, the regulatory site for GYS inactivation [5]. Phosphorylated GYS remains inactive and requires glucose 6-phosphate (G6P) for its allosteric activation. In turn, glycogen degradation requires the action of several enzymes such as glycogen phosphorylase. The final step in glycogen degradation produces glucose 6-phosphate that could be converted to glucose by G6Pase, finally appearing in the blood, to be used by other organs.

The glucagon-like peptide-1, GLP-1, is secreted by intestinal L cells in response to the presence of glucose, increasing insulin release, hence its name of “incretin”. It also promotes the proliferation of pancreatic β-cells and acts as a satiety signal in the hypothalamus [6,7,8,9]. These activities prompt its use in the treatment of type 2 diabetes and obesity, although its therapeutic role is limited by its short half-life. However, exendin-4 is a GLP-1 receptor agonist that is a more potent, stable and longer-lasting insulinotropic peptide, but with similar anorexigenic and incretin actions to GLP-1 [9,10]. Additionally, GLP-1 receptors are present in muscle [11,12] and liver [13], where GLP-1 stimulates glycogen synthesis and is a potent glycogenic hormone [14].

The PAS domain protein kinase (PASK) has been described as a nutrient sensor involved in the regulation of glucose and energy metabolism homeostasis [15,16,17,18]. PASK-deficient mice are protected against the development of obesity, insulin resistance, and hepatic steatosis when they are fed with high-fat diets (HFDs) [19,20]. We have identified PASK in hypothalamic areas and reported its role both as a nutrient sensor and as a regulator of other nutrient or energetic sensors. Thus, AMP-activated protein kinase (AMPK) and the mammalian target of rapamycin (mTOR) signaling pathways are PASK-dependent in areas involved in feeding behavior. We have also reported the role of exendin-4 modulating the activity of these nutrient sensors [21,22]. The role of PASK in insulin and glucagon secretion has already been described [23]. We also know that PASK is involved in the GLP-1 signaling pathway [21]. PASK controls glucose and lipid metabolism and critical hepatic functions regulating some of the genes and proteins responsible for glucose sensing, such as glucokinase, and for insulin signaling [24], as well as the antioxidant response and mitochondrial dynamics in the liver [25]. In addition, aged PASK-deficient mice have improved insulin sensitivity with no glucose intolerance [26]. Recently, PASK has been described as a target of mTORC1 during regenerative myogenesis in muscle stem cells [27]. PASK is a histone kinase and contributes to the methylation of H3 lysine 4 (H3K4) through association with the H3K4MLL2 methyltransferase complex [28].

As previous results have revealed PASK’s regulatory role in the responses to fasting and feeding in the liver, and the release and actions of GLP-1 are conditioned by the nutritional state, we have studied whether PASK and GLP-1 are inter-regulated and acting in unison to control glycogen storage and utilization by liver. We have therefore studied the links between exendin-4 actions and PASK in the regulation of key hepatic metabolic tasks related to glycogen storage and utilization, such as insulin signaling, glucose transport, glucose and glycogen metabolism, gluconeogenesis and de novo lipogenesis.

## 2. Materials and Methods

### 2.1. Experimental Animals and Treatments 

All procedures involving animals were approved by the appropriate Institutional Review Committee and met the guidelines for the care of animals specified by the European Community. The animals used were 10- to 16-week-old males, C57Bl/6J wild-type (WT) and PASK-deficient (*Pask*^−/−^) mice crossed into C57BL/6J for at least 12 generations [29], weighing 25–30 g. The animals were fed *ad libitum* with a standard pellet diet and housed at a constant temperature (21 °C) on a 12-h light−dark cycle, with lights on at 8 a.m.

Both PASK-deficient and WT mice were kept under standard feeding conditions (*ad libitum*) (non-fasted) or fasted for 48 h. Some animals were re-fed for three hours, and some of them were treated subcutaneously with exendin-4 (250 ng/100 g body weight, Bachem) for three hours. The mice were then decapitated, and their liver was immediately frozen or immersed in fixative for histological processing.

### 2.2. Real-Time Polymerase Chain Reaction 

The total RNA from the livers of WT and PASK-deficient mice under non-fasted, fasted and re-fed conditions was extracted with TRIzol (Life Technologies, Barcelona, Spain). RNA integrity was tested with the Bioanalyzer 2100 (Agilent), and cDNA synthesis was performed using the high-capacity cDNA archive kit (Applied Biosystems), using 2 µg of RNA as template, following the manufacturer’s instructions. Four microliters of a 1:10 dilution of the cDNA was used as a template for the polymerase chain reaction (PCR). Either TaqMan^®^ Assay (Applied Biosystems, Foster City, CA, USA) or SYBR Green^®^ Assay (Applied Biosystems) was used to quantify mRNA levels by RT-PCR in a 7300HT Fast Real-Time PCR System (Applied Biosystems). The details of the primers and probes are listed in Appendix A. The PCR conditions were 50 °C for 2 min, 95 °C for 10 min, followed by 40 cycles at 95 °C for 15 s, and 60 °C for 1 min. *18s* and *β-actin* housekeeping genes were used for normalization. In the case of SYBR Green Assay, a standard curve was previously generated in each real-time PCR assay by tenfold serial dilutions of the cDNA samples.

### 2.3. Liver Protein Detection by Western Blot

For the analysis of protein expression by Western blot, a tiny piece of frozen liver (~150 mg) was immediately lysed in a RIPA buffer (PBS, 1% NP-40, 0.5% sodium deoxycholate, 1 mM PMSF, 10 mM Leupeptin, 1 mM NA_2_VO_4_, 25 mM Na_4_P_2_O_7_, 10 mM FNa) and protease inhibitor cocktail (Roche Diagnostics, Mannheim, Germany). The tissues were immediately exposed to microwave irradiation for 5 s and then homogenized [30]. All the samples from different mice and treatments were developed in 15-well gels by SDS-PAGE. After being transferred to a PVDF membrane (Immun-Blot@ PVDF, Bio-Rad), total and activated forms of proteins were detected by Western blot using the antibodies described (Appendix A), followed by incubation with the specific secondary antibodies bound to HRP. Stain-Free Biorad© staining was used as a loading control, as previously described [20,24,25,31].

### 2.4. Assay of Glucokinase Phosphorylating Activities

Glucose-phosphorylating activities were measured using a spectrophotometric assay [32]. The activity analysis involved assays at two glucose concentrations: 0.3 mM for low-Km hexokinase (HK) activities (a concentration at which GK is essentially inactive) and 30 mM glucose (a concentration at which all phosphotransferase activities were measured). The enzyme reaction was performed using 40 μg of total protein extract in 200 μL of a reaction solution consisting of 100 mM Tris-HCl (pH 7.4), 150 mM KCl, 0.1% BSA, 1 mM MgCl_2_, 5 mM ATP-MgCl_2_, 2 mM NADP^+^ and 5.5 units of G6PDH from *Leuconostoc mesenteroides.* NADH production was measured by recording the fluorescence emitted in a spectrophotometer (Varioskan, Thermo Fisher Scientific, Waltham, MA, USA). The fluorescence values were extrapolated from a standard curve plotted with dilutions of known concentrations of NADPH. GCK activity was calculated as the difference between phosphorylating activity at high and low glucose.

### 2.5. Blood GLP-1 and Glucose Concentrations 

Blood plasma GLP-1 levels (pM) were measured by a High Sensitivity Glucagon-like peptide 1 (active) ELISA (Millipore, MA, USA), following the manufacturer’s instructions. On other hand, tail vein blood was sampled for measuring glucose with a Glucometer Elite meter (Bayer Corp. Elkhart, IN, USA).

### 2.6. Histological Liver Glycogen Detection by PAS (Periodic Acid-Schiff) Staining 

To visualize glycogen deposition in the liver, PAS (Periodic Acid Schiff) staining was performed. Livers were fixed in a modified alcoholic Bouin solution (saturated solution of picric acid in methanol (75% *v*/*v*) with formalin 37–40% formaldehyde (25% *v*/*v*) and glacial acetic acid (5% *v*/*v*)). They were dehydrated gradually and paraffin-embedded. Livers were sectioned at 7 μm, and afterwards deparaffinized and rehydrated. The PAS staining was performed with periodic acid and Schiff reagent. A hematoxylin solution was used as a nuclear stain. Slides were imaged on a Nikon Eclipse E600 microscope with a Nikon DXM1200F camera. 

### 2.7. Statistical Analyses 

Data are presented as mean ± standard error of the mean (SEM) for the indicated number of experiments. Statistical analysis was performed using GraphPad Prism 8.0.2 software (San Diego, CA, USA). Firstly, we checked for each variable normality by Shapiro−Wilk test. Two-tailed paired Student’s *t*-test to determine differences between two groups, and two-way analysis of variance (ANOVA) followed by Tukey post-hoc test or Fisher’s LSD test when there were more than two groups, were applied as appropriate. A *p*-value ≤ 0.05 was considered as statistically significant. 

## 3. Results

### 3.1. Exendin-4 and Long Fasting Downregulate Hepatic Pask Gene Expression 

PASK has been proposed as a nutrient sensor, so we were interested in discovering whether the *Pask* expression’s response to nutritional state is dependent on exendin-4. We therefore analyzed the *Pask* gene expression in the liver under standard feeding conditions (*ad libitum*) (non-fasted) as well as during fasting for 48 h in the presence or absence of exendin-4. *Pask* mRNA levels were significantly reduced after prolonged fasting (Figure 1A). The exendin-4 treatment downregulated *Pask* gene expression under all the conditions (Figure 1A), especially in non-fasted mice, as *Pask* expression was already almost undetectable in fasted conditions.

### 3.2. Liver GLP-1 Receptor Expressions and Blood GLP-1 Levels Are PASK-Dependent 

The mRNA levels coding to GLP-1 receptors (GLP1R) were measured under non-fasted and fasted conditions in WT and PASK-deficient mice. *Glp1r* expression was PASK-dependent and affected by nutritional status. Thus, PASK deficiency decreased the expression of *Glp1r* under standard feeding conditions (non-fasted) (Figure 1B). However, while fasting tended to decrease the expression of *Glp1r* in the WT, it did not do so in PASK-deficient mice, where fasting enhanced *Glp1r* expression (Figure 1B). This suggests that the regulation by fasting of GLP-1R expression was PASK-dependent. Blood GLP-1 concentrations were measured in non-fasted and fasted WT and PASK-deficient mice. Similar to what happened with the expression of the receptor, blood GLP-1 levels were lower in PASK-deficient mice under non-fasted conditions, although this effect was reversed under fasted conditions. The normal physiological response under prolonged fasting was a reduction in GLP-1 levels, as in the case of WT; however, the opposite effect was observed in PASK-deficient mice (Table 1). This result indicates that the fasting response of reducing blood GLP-1 levels depended on PASK functionality. 

### 3.3. Long Fasting Produces a Lower Weight Loss in PASK-Deficient Mice

Fasting prompted weight loss in WT and PASK-deficient mice. While WT mice significantly lost weight after 24 h, and especially during 48 h fasting, the weight loss among PASK-deficient mice was only significant after 48 h. It was interesting to note that PASK-deficient mice lost less weight than their WT counterparts (Figure 1C).

### 3.4. Exendin-4 Upregulated the GLUT-2 Protein Levels in a PASK-Dependent Way 

We studied the effect of exendin-4 and PASK on glucose transporter 2 (GLUT2), a key step in glucose entering or leaving hepatocytes according to glucose concentration in the blood. 

A 48-h fast slightly increased the expression of mRNA coding to GLUT2 in both WT and PASK-deficient mice (Figure 2A). *Glut2* expression tended to be significantly higher in PASK-deficient mice under non-fasted conditions (Figure 2A), and treatment with exendin-4 slightly upregulated its expression in WT mice, but not so in PASK-deficient mice (Figure 2A).

The effect of exendin-4 on the regulation of GLUT2 protein levels was also analyzed (Figure 2B,D). Exendin-4 sharply increased the amount of GLUT2 under both non-fasted and fasted conditions in WT mice, but not in PASK-deficient mice. It is noteworthy that the level of GLUT2 tends to be higher in both fasted and non-fasted untreated PASK-deficient mice compared to WT. Nevertheless, although exendin-4 was unable to increase GLUT-2 levels in PASK-deficient mice, these levels remained high even without exendin-4 (Figure 2B,D). 

Blood glucose levels were measured under fasted conditions in the absence or presence of exendin-4. Although glucose circulating levels were similar under fasting in both types of mice, treatment with exendin-4 significantly decreased blood glucose levels, and this effect was higher in PASK-deficient mice (Figure 2C).

### 3.5. Exendin-4 Regulated Liver GCK and GCKR Responses to Feeding/Fasting, and This Effect Is PASK-Dependent

Liver GCK catalyzes the phosphorylation of glucose in the first step of its metabolism. GCK then adjusts liver metabolism to blood glucose levels in order to maintain body glucose. This enzyme is transcriptionally and enzymatically regulated by fasting/feeding in the liver. Additionally, its function is conditioned by protein−protein interactions with GCKR. Three hours of refeeding upregulated *Gck* mRNA expression, although exendin-4 treatment prevented the recovery of this expression in both WT and PASK-deficient mice (Figure 3A), suggesting that the transcriptional regulation of *Gck gene* was PASK-independent. However, an analysis of the effect of exendin-4 treatment on GCK and GCKR protein expression revealed the important role PASK plays. Thus, the presence of exendin-4 for three hours decreased the level of GCK protein under basal conditions in WT mice, but not in PASK-deficient mice. Exendin-4 treatment also reversed the fasting effect in PASK-deficient mice, maintaining similar levels of GCK in fasting and basal states (Figure 3C,E). However, exendin-4 did not significantly modify the levels of GCKR in either WT or PASK-deficient mice (Figure 3D,E).

The effect of exendin-4 on GCK activity was also tested. Three hours of treatment with exendin-4 in refeeding mice significantly decreased liver total phosphorylating activities in both WT and PASK-deficient mice (Figure 3F). We analyzed GCK activity independently to rule out a different contribution by GCK or HK (hexokinase) to total glucose phosphorylating activity in both mice. GCK’s contribution to total glucose phosphorylating activity was similar at ~70% in PASK-deficient and WT mice (Figure 3G), suggesting that PASK is not essential in the exendin-4 inhibitory action on GCK activity. 

### 3.6. Exendin-4 and PASK Regulate Glycogen Stores in the Liver in Response to Long Fasting

We analyzed the effect of exendin-4 and PASK deficiency on glycogen storage in the liver from non-fasted and 48-h fasted WT and PASK-deficient mice in the presence or absence of exendin-4 (Figure 4). Periodic Acid-Schiff (PAS) stain was used accordingly to detect polysaccharides such as glycogen depots in hepatocytes from liver tissue sections. A majority presence of PAS-positive cells was observed under non-fasted conditions, indicating a high content of glycogen in their cytoplasm in both WT and PASK-deficient livers. PAS-positive cells with high glycogen depots were found mainly around the portal area (Figure 4a,e). Three hours of exendin-4 treatment also involved an accumulation of large glycogen deposits, although in an uneven pattern (Figure 4b,f). There were no glycogen depots in livers from fasted WT mice, although they had more lipid droplets (Figure 4c). By contrast, fasting reduced hepatocyte glycogen depots in PASK-deficient mice, although some of them retained small amounts, indicating a slightly higher glycogen presence and decreased lipid droplets compared to fasted WT mice (Figure 4g). Exendin-4 treatment increased the number of PAS-positive glycogen depots detected under fasted conditions in WT mice and decreased lipid droplets (Figure 4d). However, exendin-4 treatment did not significantly increase glycogen accumulation in livers from fasted PASK-deficient mice, and their hepatocytes had numerous small lipid droplets in their cytoplasm (Figure 4h).

### 3.7. The Response of Glycogen-Metabolism Associated Proteins to Long Fasting Was Modified by Exendin-4 and Altered by PASK Deficiency

We analyzed some of the key enzymes in hepatic glycogen synthesis (Glycogen synthase, GYS) and utilization (glycogen phosphorylase liver (PYGL)). Additionally, GYS has been identified as one of the substrates of PASK. We analyzed the effect of exendin-4 and PASK on gene expression of these genes. mRNA levels were measured in the liver from non-fasted and 48-h fasted WT and PASK-deficient mice in the presence or absence of exendin-4. 

Glycogen synthase expression under non-fasted conditions was slightly higher in PASK-deficient mice than in WT. However, prolonged fasting (48 h) increased *Gys*2 mRNA expression in WT mice but did not alter its expression in PASK-deficient mice (Figure 5A). Nevertheless, the effect of exendin-4 treatment was observed only under fasted conditions in PASK-deficient mice, increasing *Gys2* expression (Figure 5A). These data indicate that both the fasting response and the response to exendin-4 in that condition were PASK-dependent.

The effect of exendin-4 on the expression of protein GYS and phospho-GYS (inactive forms) was also analyzed (Figure 5B,D). Exendin-4 increased the amount of GYS under both non-fasted and fasted conditions in WT mice, while prolonged fasting decreased the amount of GYS (Figure 5B,D). However, GYS was overexpressed in PASK-deficient mice, especially under fasted conditions, in contrast to the WT response. As happened with GLUT-2, exendin-4 had no effect in PASK-deficient mice, probably because the amount of GYS proteins was already high in these mice, especially in fasting conditions (Figure 5B,D). 

The phosphorylation of GYS maintains this complex as inactive. Several reversible phosphorylation sites were identified, with the C-terminal sites being the main regulatory ones. We used anti-phospho-GYS (Ser 641) (P-GYS), which recognizes phospho Ser 640 and 641, both phosphorylated by GSK3 and PASK. Prolonged fasting increased the inactive form of GYS (P-GYS) in WT mice (Figure 5C,D). However, the presence of exendin-4 blocked it. P-GYS was always lower in PASK-deficient mice, with no observable response to exendin-4 treatment (Figure 5C,D), indicating higher levels of active enzyme, and therefore higher glycogen synthesis.

Glycogen phosphorylase liver (*Pygl*) expression was similar under both non-fasted and fasted conditions in WT and PASK-deficient mice (Figure 5E). Exendin-4 treatment tends to overexpress glycogen phosphorylase mRNA under non-fasted conditions, being significant only in PASK-deficient mice (Figure 5E).

### 3.8. Exendin-4 Regulated AKT Activity Differently in the Basal State or after Fasting or Refeeding States, and It Is a PASK-Dependent Effect

Insulin signaling initially involves the autophosphorylation of the tyrosine residues of the insulin receptor, generating docking sites for signaling proteins. The activation of phosphatidylinositol-3,4,5-triphosphate kinase (PI3K) initiates the metabolic signaling pathway. Phosphoinositide-dependent kinase-1 (PDK1) activation induces the partial activation of AKT, and full activation requires Ser473 phosphorylation by other kinases (probably mTORC2), finally regulating multiple substrates. In addition, insulin/PI3K signaling depends on the phosphatase PTEN, which is a widely-known negative regulator. We analyzed the exendin-4 effect on the expression and activation levels of AKT in the liver from non-fasted and 48-h fasted WT and PASK-deficient mice, and after three hours of refeeding (Figure 6). 

Exendin-4 inhibited the AKT activation in non-fasting conditions with similar levels to the very low activation found in fasted or short-refed WT mice. Exendin-4 thus mimics the response to fasting in AKT activation, with clear inhibitory effects exclusively when AKT was active. However, the over-activation of AKT observed under fasted and refed conditions in PASK-deficient mice was not reversed by exendin-4 treatment (Figure 6B), indicating that both the response to fasting and the actions of GLP-1 in these conditions are dependent on PASK.

The PI3K/Akt pathway is also regulated by PTEN, which controls the phosphorylation state of PIP_3_, PI3K and AKT. We tested the effect of exendin-4 treatment on the levels of hepatic PTEN, revealing that exendin-4 treatment decreased PTEN protein levels in non-fasted WT mice, but not in PASK-deficient ones (Figure 6C). However, exendin-4 treatment increased PTEN levels significantly after refeeding in both WT and PASK-deficient mice.

### 3.9. Exendin-4-Modulated Expression of Hepatic Metabolic Genes in Response to Fasting and Refeeding, and Some of These Effects Are PASK-Dependent

We evaluated the role of exendin-4 and PASK in hepatic function by analyzing the expression of the transcription factors and genes that regulate the main metabolic pathways in the liver (Appendix A): glycolysis, lipogenesis and gluconeogenesis. mRNA levels were measured in the liver from non-fasted and 48-h fasted WT and PASK-deficient mice in the presence or absence of exendin-4. The genes stimulated by feeding are included in Appendix A (transcription factors: *Lxrα*, *Srebp1c* and *Chrebp*; glycolysis enzymes: *L-pk*; lipogenic enzymes: *Fas* and *Scd1)* while the genes activated under fasting conditions are included in Appendix A (transcription factors: *Foxo1, Ppargc1a, Pparα* and *Pparγ*; transcription coactivators: *Sirt1* and *Sirt2* in Appendix A; gluconeogenic enzymes: *Pepck* and *G6pase*; fatty acid metabolism: *Cpt1a* and *Mcad* in Appendix A).

Exendin-4 treatment in the basal state decreased several genes induced by feeding: *Srebp1c, L-pk, Fas and Scd1* in WT mice, and the same or greater effect was observed in PASK-deficient mice. The inhibitory effect of exendin-4 was only observed in *Lxrα* and *Chrebp,* in PASK-deficient mice (Appendix A). 

When the exendin-4 effect was analyzed under fasted conditions, it also reversed the upregulating-fasting effect on the expression of several transcription factors and coactivators, such as the following: *Pparα**, Sirt2,* and genes that regulate the main metabolic pathways in the liver, such as *G6pase* and *Cpt1a* and slightly *Mcad* (Appendix A). Conversely, exendin-4 stimulated *Ppargc1a* and *Pparγ* expression. Differences in the exendin-4 effect under fasted conditions were also observed between WT and PASK-deficient mice, such as the inhibition of the expression of *Foxo1, Sirt1* and *Pepck,* while a slight increase of *Mcad* genes was only observed in PASK-deficient mice, while exendin-4 treatment decreased the expression of *Cpt1a* and slightly *Mcad* only in WT mice (Appendix A). 

## 4. Discussion

Glucagon-like peptide-1 (GLP-1) is an incretin hormone that increases insulin secretion and inhibits glucagon, maintaining glucose homeostasis [9], and it has anorectic properties [6,7]. This has led to the use of exendin-4 (an analogue of GLP-1 with a longer half-life) in the clinical treatment of type 2 diabetes [9,10]. Furthermore, other GLP-1 receptor agonists, such as oral semaglutide (sema), are now being developed for the treatment of type 2 diabetes [33]. 

Another controller of homeostasis and energy metabolism is PASK [16,19]. Its alteration has been related to the development of type 2 diabetes [23,34]. PASK is also a hypothalamic nutrient and energy sensor [21], controlling the development of obesity under an HFD. In fact, our previous results indicate that PASK deficiency reverses most of the deleterious changes induced by an HFD in relation to liver fasting/feeding adjustment. This involves highly regulated molecular mechanisms that control the expression and function of the transcription factors, enzymes and miRNAs in glucose and insulin signaling [20,24]. Pharmacologic inhibition of PASK also confirmed its capacity for improving insulin sensitivity, reducing nonalcoholic steatohepatitis caused by an HFD [35]. Accordingly, PASK has also been proposed as a possible target in the treatment of diabetes and obesity. 

The liver is a vital organ for maintaining metabolic homeostasis, providing the metabolic adjustment required in fasting and feeding periods. Insulin signaling and glycogen metabolism therefore play a crucial role. Reduced hepatic insulin sensitivity involves postprandial hyperglycemia through hepatic glucose overproduction, and consequently chronic hyperinsulinemia that triggers type 2 diabetes [36,37,38,39]. This deregulation also promotes lipogenesis, inducing hepatic steatosis and further systemic insulin resistance [40].

We have therefore investigated the interrelationships between exendin-4 and PASK in insulin response, glucose transport, glycogen metabolism and de novo lipogenesis in order to adjust the liver’s role in nutritional changes to fasting and postprandial conditions.

In previous studies, we have described how PASK plays an important role in GLP-1′s actions in neuroblastoma N2A cells [22] and in the hypothalamic areas involved in feeding behavior [21]. Additionally, GLP-1/Exendin-4 reverses glucose’s effects on other metabolic sensors and energy regulators such as AMPK and p70S6K, and modulates its activity in metabolic alterations such as obesity or insulin resistance [21,30]. 

In this report, we first analyzed the effect of exendin-4 on hepatic *Pask* expression and the impact of PASK deficiency on blood GLP-1 concentration and GLP-1 receptor expression to understand their role in the adjustment to fasting/feeding conditions. 

Our data show that a three-hour treatment with exendin-4 blocks *Pask* expression under both fasting and feeding conditions. This effect was similar to the one described in the lateral hypothalamic (LH) area [21], although the exendin-4 effect, in this latter case, was significant only under fasting conditions. Our data suggest that PASK and GLP-1 are performing mutual inter-regulation, and some of the effects of exendin-4 might be mediated by PASK, possibly affecting other liver metabolic sensors and regulators, as several hepatic functions are PASK-dependent [18,19,20,24,25,41]. 

Thus, both the hepatic expression of *Glp1r* and plasma GLP-1 concentrations are PASK-dependent. Furthermore, *Pask* expression is controlled by GLP-1. In fact, PASK deficiency disrupted the normal physiological GLP-1 response to fasting. Previous studies indicate that heterozygous *Glp1r* mice have altered glycemia and, consequently, reduced blood insulin levels in response to glucose, but their glucagon levels are unaffected, and GLP-1 levels are similar in both (*Glp1r*^+/−^) and (*Glp1r*^−/−^) mice [42]. Nevertheless, we have previously described how one hour of exendin-4 treatment did not alter circulating insulin levels in WT or PASK-deficient mice under either fasting or feeding conditions [21]. The insulin regulatory effect on hepatic glucose metabolism might also depend on insulin actions in other organs [43,44]. Likewise, GLP-1 glucoregulatory control also seems to be mediated by a neural effect [45]. 

Feeding increases blood glucose, and insulin release promotes the expression of LXRα and SREBP1c, so glucose induces the expression and activation of CHREBP [3,46]. They all promote the transcriptional activation of glucose transporter 2 as well as glycolytic and lipogenic enzymes [3,47,48]. 

Exendin-4 is reported to increase GLUT2 coding mRNA and protein expression in a pancreatic beta-cell line [49]. In contrast, our data indicate that a short exendin-4 treatment and PASK deficiency slightly increases hepatic *Glut2* mRNA, but after this treatment, the protein GLUT2 increased only in WT, as PASK-deficient mice have already increased the protein per se, and exendin-4 has no enhancing effect, probably due to a previous saturation of the response. Exendin-4 and PASK deficiency might therefore facilitate glucose entering and exiting hepatocytes.

Hepatic GCK is crucial for glycogen synthesis and storage, and therefore for fasting and feeding responses. Its expression and activity are regulated by transcriptional and post-transcriptional mechanisms [50]. For instance, hepatic GCK gene expression is insulin-dependent [51], and its activity is regulated by GCKR. The result is that GCK expression is reduced during fasting, as is its activity, because GCK and GCKR are bound in the nucleus. After refeeding, GCK is released from GCKR and translocated to the cytoplasm, ready to phosphorylate glucose [4,52]. We have previously reported a faster recovery of mRNA, coding to GCK and GCKR, in refed PASK-deficient mice after a long fasting period [24]. 

Our data show that exendin-4 affects the insulin signaling pathway—the main transcriptional regulator of GCK expression. Exendin-4 treatment blocks AKT activation under non-fasted conditions in WT and PASK-deficient mice. We have already described how AKT was overactivated in PASK-deficient mice [24]. Nevertheless, the exendin-4 effect blocking AKT activation was also lost in fasted and refed PASK-deficient mice. The exendin-4 effect blocked the expression of *Lxrα*, *Srebp1c* in WT and PASK-deficient mice. Additionally, this effect might still be greater on PASK deficiency by also decreasing *Chrebp*. CHREBP^-/-^ mice recorded a lower liver lipogenic gene expression and triglyceride levels, while a compensatory increase in glycogen stores was observed [53]. Exendin-4 treatment disabled the *Gck* expression induced by refeeding. Exendin-4 treatment also decreased GCK protein levels in WT mice under basal conditions. In contrast, PASK deficiency reversed this effect, as GCK expression was always higher after exendin-4 treatment under all conditions, and especially under fasting, when GCK expression was lower in step with higher levels of GCK. 

Glucokinase is the key rate-limiting enzyme of hepatic glycolysis, and it enhances the flow of glucose via glycolysis and increases Acetyl-CoA production for de novo lipogenesis. The lower activity in response to exendin-4 treatment should also reduce de novo lipogenesis in accordance with the lower expression of *L-pk*, *Fas* and *Scd1*. Our data are consistent with those reported for the cultured hepatocytes of mice maintained ad libitum, where exendin-4 decreases the expression of the transcription factors and enzymes involved in de novo lipogenesis, such as *Srebp1c*, *Acc* and *Scd1* [54]. Our results suggest that some of the hepatic effects of exendin-4 are PASK-dependent and could be more effective in PASK-deficient mice. 

Glycogen metabolism in the liver helps to regulate the level of blood glucose. It involves the antagonistic responses of glycogen synthase (GYS2) and phosphorylase (PYGL) activity. Thus, the signals that promote glycogen degradation inhibit its synthesis, and vice versa. Moreover, the inhibition of G6Pase activity has previously been related to an increase in glycogen synthesis under fasted conditions [55]. 

An increase in the postprandial level of glucose drives the dephosphorylation and inactivation of the glycogen phosphorylase (PYGL, which is responsible for glycogen degradation) [4]. Conversely, the same protein phosphatase activates hepatic glycogen synthase (GYS2) by dephosphorylating, and so glycogen is synthetized. 

PASK phosphorylates and regulates glycogen synthase activity [5] through the phosphorylation of the Ser 640 site, which has also been identified as a substrate of GSK3. However, the role of PASK in liver glycogen synthesis and storage remains unknown. We have analyzed the roles of PASK and exendin-4 in glycogen metabolism and storage, focusing on glycogen phosphorylase and synthase expressions. 

The activity of glycogen phosphorylase and synthase in the liver changes in minutes in a glucose-dependent way, due mainly to phosphorylation/dephosphorylation. Our results show that the glycogen phosphorylase expression was not modified in the adjustment to fasting/feeding, nor was it PASK-dependent, but exendin-4 upregulated the glycogen phosphorylase expression under non-fasted conditions. Nevertheless, the glycogen synthase expression was modulated by fasting, exendin-4 and PASK, and the exendin-4 effect is PASK-dependent. Prolonged fasting therefore unexpectedly induced *Gys2* expression in WT, and slightly so in PASK-deficient mice. This effect was specific to prolonged fasting (48 h), as 24-h fasting did not modify *Gys2* expression (data not shown). The presence of exendin-4 increases *Gys2* expression under prolonged fasted conditions in PASK-deficient mice. However, the response to fasting and feeding agrees with previous data, when glycogen synthase activity and protein levels are checked, as occurs in fed and fasted states in normal rats [56].

Another interesting finding is that exendin-4 increases GYS protein expression in WT mice. This effect might be mediated by increasing translation more so than at the transcriptional level, in keeping with the differences in mRNA and protein expression previously reported [56,57]. GYS protein was overexpressed in PASK-deficient mice, especially under fasted conditions, and exendin-4 treatment maintained the higher expression. The level of phospho-GYS in Ser 640/641 indicates an inactive enzyme, and therefore a lack of new glycogen storage. Our results show a strong phosphorylation state (inactivity of the enzyme) under fasted conditions in WT mice, confirming that GYS expression was low and inactive under those conditions. These data are also supported by the absence of glycogen depots in the hepatocytes observed by PAS staining. Our data also showed that the phosphorylation of liver GYS2 at Ser 640/641, and therefore the inactivation of this enzyme, was strongly blocked in PASK-deficient mice. It is in accordance with the in vitro finding of the muscle glycogen synthase (GYS1) inactivation by direct PASK phosphorylation at Ser 641 [5], suggesting also a physiological role of PASK in muscle glycogen synthesis as we have shown in the liver here. In addition to Ser 640/641, other phosphorylations on N-terminal residues could also be participating in GYS inactivation, as previously proposed by [58]. Interestingly, three hours of exendin-4 treatment blocked the phosphorylation of Ser640/641, suggesting that this could activate GYS and initiate glycogen storage in hepatocytes, as evidenced by PAS staining, through the presence of small particles of glycogen deposited on hepatocytes. This response can be explained by the exendin-4 effects previously described in this report: inhibiting AKT activity and blocking *Pask* expression. PASK-deficient mice always maintained lower GYS phosphorylation than WT mice both in fasted and fed mice, and exendin-4 treatment did not have any additional effect, probably due to low glucose availability, as *Pepck* expression is also inhibited. The low phosphorylation of GYS in PASK-deficient mice could also be explained by a combination of the lack of phosphorylating actions of PASK (in a direct manner) and GSK3β (indirectly), as PASK deficiency activates AKT and therefore keeps GSK3 inactive. Defective muscle glycogen synthesis has been reported in type 2 diabetes patients [59]. Phosphorylation of muscle glycogen synthase by PASK [5] or GSK3 is in addition the main regulatory modification for its catalytic activity [60]. Dephosphorylation of this site increased glycogen synthesis [61]. 

Under fasting conditions, when blood glucose concentration decreases, liver gluconeogenesis is activated, and glucose is exported to the blood. By contrast, glycolysis decreases under these conditions, and fatty acids become the fuel the liver uses to generate energy. Our previous results show that prolonged fasting increases the expression of the hepatic genes involved in the activation of gluconeogenesis, the output of glucose to the blood, and the increased β-oxidation of fatty acids [24]. In turn, PASK-deficient mice recorded a significant decrease in the level of expression of several genes induced by fasting: *Foxo1, G6pase, Cpt-1a* and *Mcad* [24,25], although no differences in glycemic control were detected between the two types of mice after 24-h and 48-h fasting periods [21]. Prolonged fasting causes less weight loss in PASK-deficient mice, which suggests that energy stores are better maintained.

Our results show that the presence of exendin-4 under fasting conditions increases *Ppargc1α* expression, while *Foxo1* and *Sirt1* expression decreases in PASK-deficient mice. The expression of the gluconeogenesis enzymes PEPCK and G6Pase also depends on insulin signaling, which regulates FOXO1 acetylation, and consequently phosphorylation, increasing its cytosolic location [62,63]. Exendin-4 only reversed AKT overactivation under non-fasted conditions, with AKT activity still being elevated under fasted conditions in PASK-deficient mice. This might explain the lower expression of *Pepck* in PASK-deficient mice. *G6pase* expression was blocked in both WT and PASK-deficient mice, which could explain the rapid decrease in blood glucose observed in both types of mice [21], as G6Pase is critical for allowing the output of glucose from the cell to the circulatory torrent. In contrast, although exendin-4 decreases *Pparα* expression, the effect was different in *Cpt1α* expression, which decreases in WT mice, while this effect was not observed in PASK-deficient mice. The exendin-4 effect on *Mcad* expression was also PASK-dependent. While a slight decrease was detected in WT, a slight increase was observed in PASK-deficient mice, which might explain the excess of lipid droplets found in fasted WT mice, whereby an increase in lipid droplets correlates with diminished mitochondrial fatty acid transport and lipolysis in WT mice.

## 5. Conclusions

In short, our data show that the effects of exendin-4 on the liver’s adjustment to fasting/feeding could be conditioned by PASK activity, and PASK inactivation could be enhancing those effects. We therefore describe for the first time how exendin-4 regulates the expression of the hepatic *Pask* gene, and how PASK modulates *Glp1r* expression and blood GLP1 levels. In a non-fasted state, exendin-4 blocks hepatic AKT activation, *Lxrα Srebp1c* and both gene expression and GCK activity, so the expression of *L-pk* and the lipogenic genes is impaired after a three-hour treatment, while GLUT2 and GYS are increased. PASK deficiency increases some of the inhibitory effects of exendin-4, also decreasing *Chrebp* expression. Furthermore, *G6pase* expression is blocked under fasted conditions in both WT and PASK-deficient mice, while GYS expression increases, especially in the presence of exendin-4 in PASK-deficient mice. More glycogen depots are detected under fasted conditions by exendin-4 or by PASK deficiency. Our data suggest that exendin-4 might improve its efficiency when PASK activity is impaired in parallel, as occurs in PASK-deficient mice. Some deleterious consequences of type 2 diabetes and HFDs could therefore be reverted by exendin-4 and PASK deficiency and/or inactivation through the regulation of glucose transport and glycogen metabolism, in addition to the lipid metabolism already described.

## Figures and Tables

**Figure 1 nutrients-13-02552-f001:**
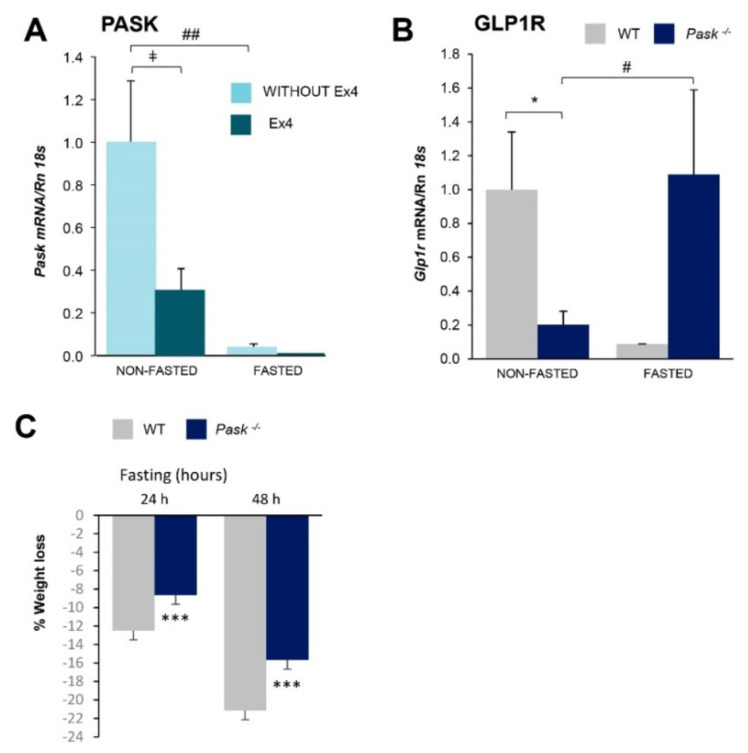
Exendin-4 regulates *Pask* gene expression in the liver and the expression of GLP-1 receptors, and serum GLP-1 levels are PASK-dependent. Quantitative real-time PCR was used to analyze the expression *Pask* mRNA level in liver from wild-type mice (**A**) were measured in WT non-fasted (NON-FASTED) and 48-h fasted (FASTED) specimens in the absence or presence of exendin-4 (Ex4) treatment, and the *Glp1r* mRNA from WT and PASK-deficient mice were measured in non-fasted (NON-FASTED) and 48-h fasted (FASTED) specimens (**B**). *Pask* mRNA levels (**A**) and *Glp1r* mRNA levels (**B**) were normalized by Rn *18s*. Results are shown as mean ± SEM of 3–5 animals per condition and expressed as fold-over the non-fasted condition mean; ^#^
*p* < 0.05, ^##^
*p*< 0.01 non-fasted vs. 48-h fasted, ^‡^
*p* < 0.05 vehicle vs. exendin-4 and * *p* < 0.05 WT vs. *Pask*^−/−^ by two-way ANOVA followed by Tukey post-hoc test. Body weights were recorded in WT and PASK-deficient mice before and after 24 or 48 h of fasting (**C**). The percentage of weight lost was represented in the graph, where bars represent the mean ± SEM of 6–8 animals per group; *** *p* < 0.001 WT vs. *Pask*^−/−^ by paired *t*-test.

**Figure 2 nutrients-13-02552-f002:**
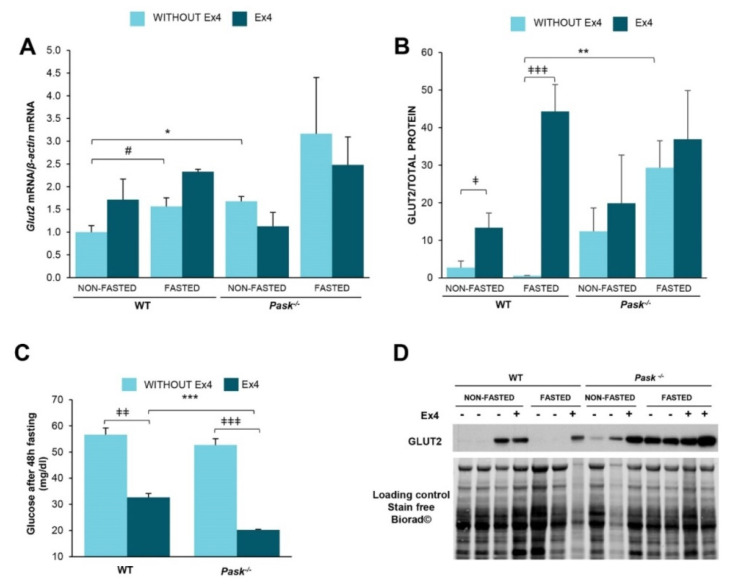
GLUT2 gene expression, protein expression and blood glucose levels altered by exendin-4 and PASK deficiency. Quantitative real-time PCR was used to analyze the expression of *Glut2* (**A**). mRNA levels were measured in non-fasted (NON-FASTED) and 48-h fasted (FASTED) liver from wild-type mice (WT) and PASK-deficient mice (*Pask*^−/−^) treated or not with exendin-4 (Ex4). mRNA levels were normalized by mRNA of *β-actin* used as a housekeeping gene. Immunoblot analysis of total GLUT2 (GLUT2) (**B**,**D**) in liver from wild-type (WT) and PASK-deficient mice (*Pask*^−/−^). Liver lysates from non-fasted (NON-FASTED) and 48-h fasted (FASTED) specimens in the presence or absence of exendin-4 (Ex4) were processed for Western blot analysis. A representative blot of 2 prepared blots is also shown (**D**), from which the data shown in the graph were collected (**B**). The values were normalized by total protein detected by Stain-Free Biorad© and expressed as fold-over the non-fasted wild-type mice data are shown as mean ± SEM of 3–5 animals per condition; ^‡^
*p* < 0.05, ^‡‡‡^
*p* < 0.001 vehicle vs. exendin-4; * *p* < 0.05; ** *p* < 0.01 WT vs. *Pask*^−/−^ and ^#^
*p* < 0.05 non-fasted vs. 48-h fasted by two-way ANOVA followed by Tukey post-hoc test. Blood glucose levels (mg/dl) in fasting mice (**C**) were measured in wild-type mice (WT) and PASK-deficient mice (*Pask*^−/−^) treated or not with exendin-4 (Ex4). Bar graphs show mean ± SEM of 6–8 animals per group; ^‡‡^
*p* < 0.01, ^‡‡‡^
*p* < 0.001 vehicle vs. exendin-4; *** *p* < 0.001and WT vs. *Pask*^−/−^ by two-way ANOVA followed by Tukey post-hoc test.

**Figure 3 nutrients-13-02552-f003:**
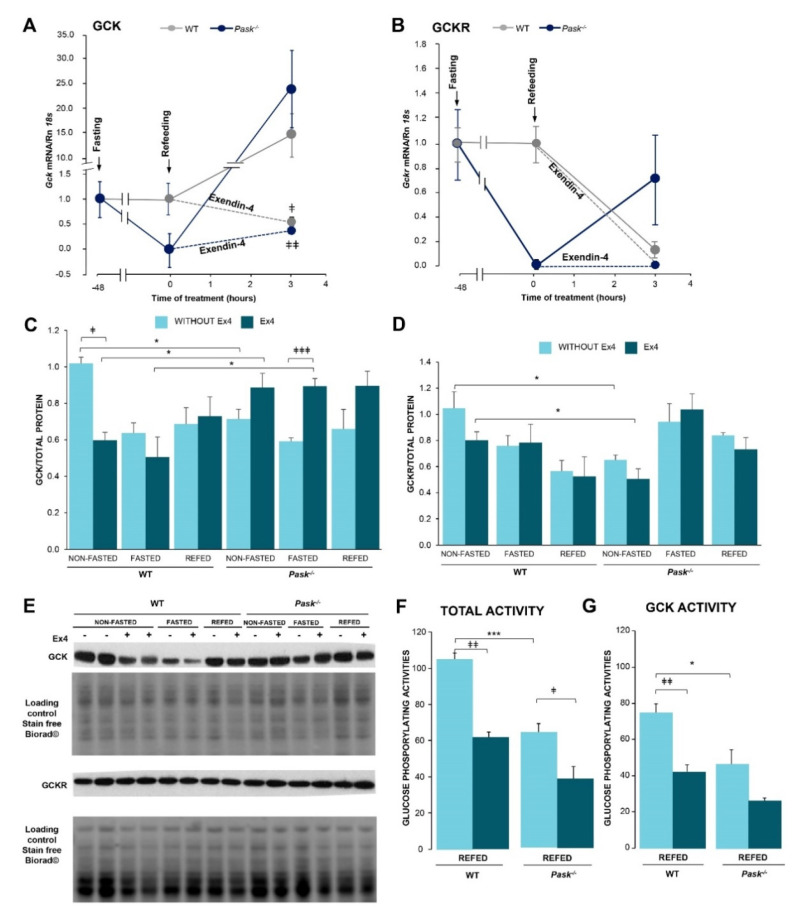
Effects of exendin-4 and PASK deficiency on the response of hepatic *Gck, Gckr* mRNAs and GCK, GCKR protein levels to fasting/refeeding and glucokinase activity. Quantitative real-time PCR was used to analyze the expression of *Gck* (**A**) and *Gckr* (**B**) mRNA levels in the liver from wild-type (WT) and PASK-deficient mice (*Pask*^−/−^). mRNA levels were measured in 48-h fasted (FASTED) and 3-h refed (REFED) specimens in the absence or presence of exendin-4 treatment and normalized by *Rn 18s*. Data are shown as mean ± SEM of 3–5 animals per group and expressed as fold-over the fasted condition mean. A *Gck* mRNA: ^‡^
*p* < 0.05, ^‡‡^
*p* < 0.01 vehicle vs. exendin-4 by two-way ANOVA followed by Fisher’s LSD test. Immunoblot analysis of GCK (**C**,**E**) and GCKR (**D**,**E**) in the liver from wild-type (WT) and PASK-deficient mice (*Pask*^−/−^). Liver lysates from non-fasted (NON-FASTED), 48-h fasted (FASTED) and 3-h refed (REFED) livers in the absence or presence of exendin-4 (Ex4) treatment were processed for Western blot analysis. A representative blot of 2 prepared blots is also shown (**E**), from which the data shown in the graph were collected (**C**,**D**). The values were normalized by total protein detected by Stain-Free Biorad© and expressed as fold-over the non-fasted wild-type mice. Data are shown as mean ± SEM of 3–5 animals per condition; * *p* < 0.05 WT vs. *Pask*^−/−^, ^‡^
*p* < 0.05, ^‡‡‡^
*p* < 0.001 vehicle vs. exendin-4 by two-way ANOVA followed by Tukey post-hoc test. Total glucose-phosphorylating activity assays (**F**) were performed at 30 mM, with low-Km hexokinase (HK) activities at 0.3 mM glucose. GCK activity was obtained by the difference between individual values at high and low glucose (**G**). Liver homogenates from 3-h refeeding after prolonged fasting (REFED) in the absence or presence of exendin-4 (Ex4) treatment from wild-type (WT) and PASK-deficient mice (*Pask*^−/−^). (**F**) Total glucose-phosphorylating activity and (**G**) glucokinase (GCK) activities are shown as mean ± SEM of 3–5 animals per group and expressed as a percentage of the control mean; * *p* < 0.05, *** *p* < 0.001 WT vs. *Pask*^−/−^ and ^‡^
*p* < 0.05, ^‡‡‡^
*p* < 0.001 vehicle vs. exendin-4 by two-way ANOVA followed by Tukey post-hoc test.

**Figure 4 nutrients-13-02552-f004:**
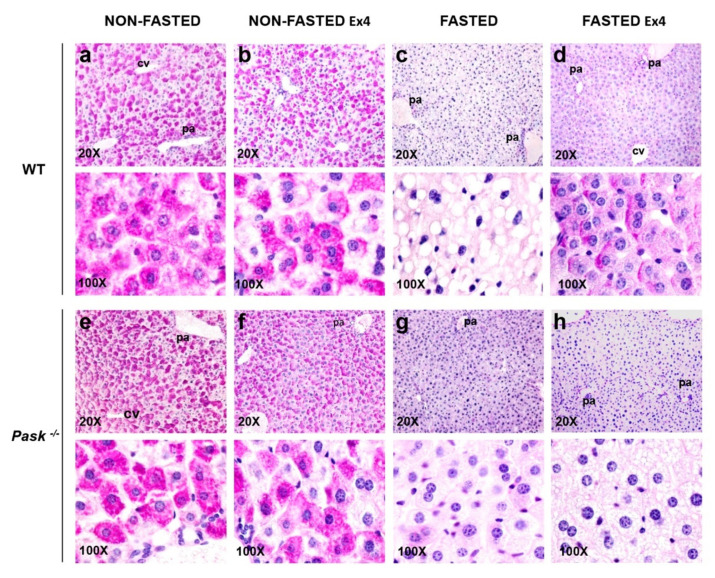
Exendin-4 and PASK deficiency alter glycogen accumulation in the liver. Histological sections from livers of wild-type (WT) and PASK-deficient mice (*Pask*^−/−^) either non-fasted (NON-FASTED) or 48-h fasted (FASTED), treated or not with exendin-4 (Ex4) for 3 h, were stained with PAS for glycogen examination. Representative micrographs of WT (**a**–**d**) and PASK-deficient mice (**e**–**h**) are shown at 20× (upper panels) and 100× magnification (bottom panels). Two different mice were used in each condition. pa: portal area, cv: centrilobular vein.

**Figure 5 nutrients-13-02552-f005:**
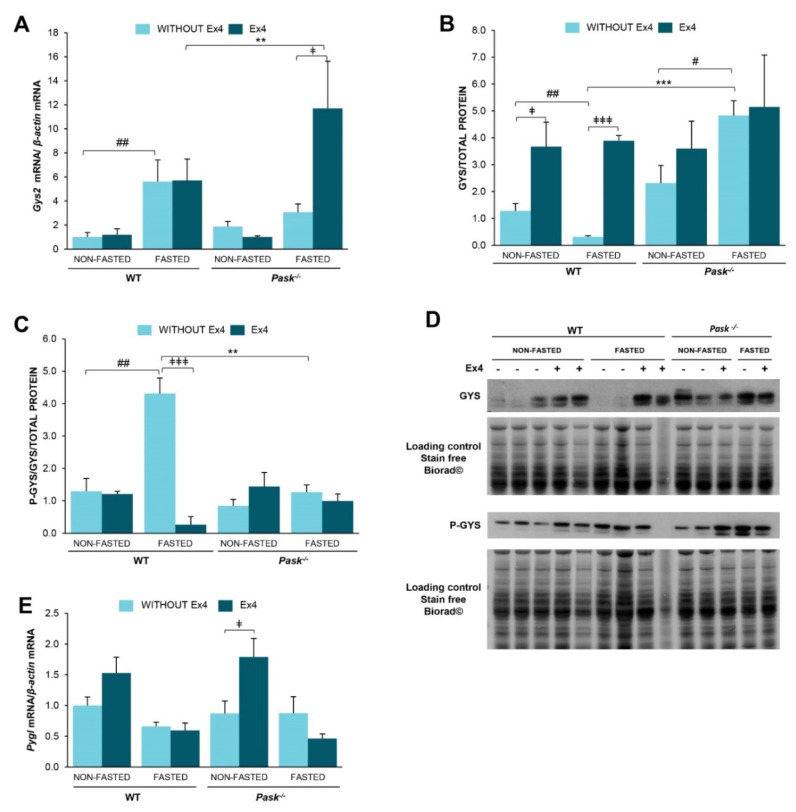
Exendin-4 and PASK deficiency alter *Gys2* and *Pygl* gene expression as well as GYS protein expression and P-GYS. Quantitative real-time PCR was used to analyze the expression of glycogen synthase *Gys2* (**A**) and glycogen phosphorylase *Pygl* (**E**). mRNA levels were measured in non-fasted (NON-FASTED) and 48-h fasted (FASTED) livers from wild-type mice (WT) and PASK-deficient mice (*Pask*^−/−^) treated or not with exendin-4 (Ex4). The mRNA levels were normalized by mRNA of *β-actin* used as housekeeping gene. Data are shown as mean ± SEM of 3–5 animals per group and expressed as fold-over the non-fasted wild-type mice of 3–5 animals per condition (** *p* < 0.01 WT vs. *Pask*^−/−^, ^##^
*p* < 0.01 non-fasted vs. 48-h fasted, ^‡^
*p* < 0.05 vehicle vs. exendin-4 by two-way ANOVA followed by Tukey post-hoc test). Immunoblot analysis of total Glycogen synthase (GYS) (**B**,**D**) and the level of phospho-GYS (P-GYS) protein expression (**C**,**D**) in the liver from wild-type (WT) and PASK-deficient mice (*Pask*^−/−^). Liver lysates from non-fasted (NON-FASTED) and 48-h fasted (FASTED) specimens in the presence or absence of exendin-4 (Ex4) were processed for Western blot analysis. A representative blot of 2 prepared blots is also shown (**D**), from which the data shown in the graph were collected (**B**,**C**). The values were normalized by total protein detected by Stain-Free Biorad© and expressed as fold-over the non-fasted wild-type mice. Data are shown as mean ± SEM of 3–5 animals per condition and expressed by fold-over the non-fasted wild-type mice. ** *p* < 0.01, *** *p* < 0.001 WT vs. *Pask*^−/−^, ^#^
*p* <0.05, ^##^
*p* <0.01 non-fasted vs. 48-h fasted and ^‡^
*p* < 0.05, ^‡‡‡^
*p* < 0.001 vehicle vs. exendin-4 by two-way ANOVA followed by Tukey post-hoc test.

**Figure 6 nutrients-13-02552-f006:**
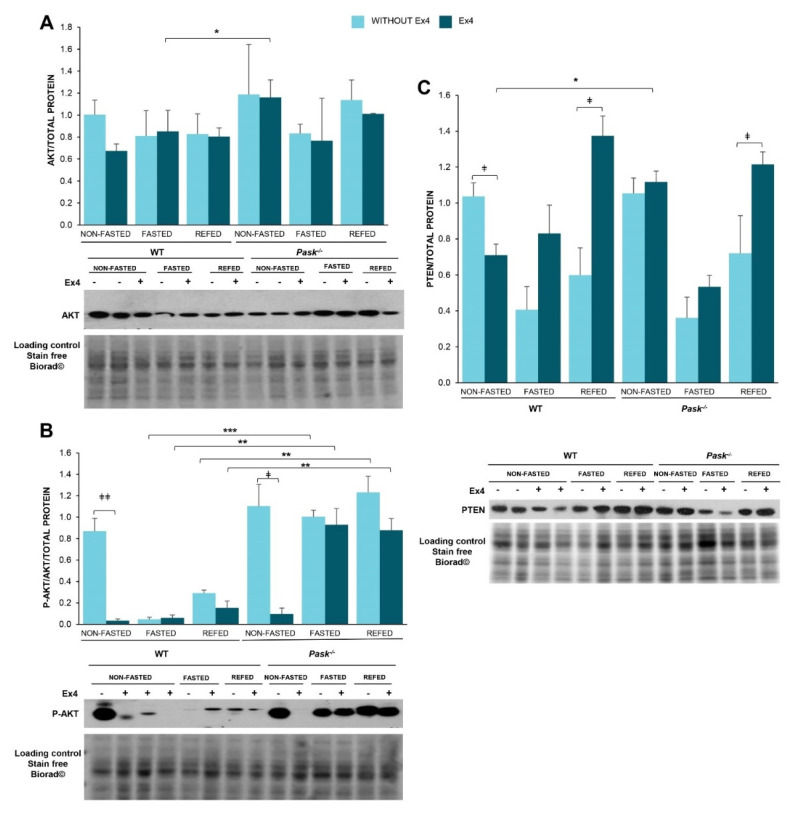
Exendin-4 and PASK-deficiency alter the adjustment of the insulin signaling pathway to fasting or refeeding states. Immunoblot analysis of phospho-AKT (Ser 473) (P-AKT) and total AKT (AKT) (**A**,**B**) as well as total PTEN (PTEN) (**C**) in the liver from wild-type (WT) and PASK-deficient mice (*Pask*^−/−^). Liver lysates from non-fasted (NON-FASTED), 48-h fasted (FASTED) and 3-h refed (REFED) specimens in the presence or absence of exendin-4 (Ex4) were processed for Western blot analysis. The values were normalized by total protein detected by Stain-Free Biorad© and expressed as fold-over the non-fasted wild-type mice. A representative blot of 2 prepared blots is also shown, from which the data shown in the graph were collected. Bar graphs show mean ± SEM of 3–5 animals per condition and expressed by fold-over the non-fasted wild-type mice; * *p* < 0.05, ** *p* < 0.01, *** *p* < 0.001 WT vs. *Pask*^−/−^ and ^‡^ *p* < 0.05, ^‡‡^ *p* < 0.01 vehicle vs. exendin-4 by two-way ANOVA followed by Tukey post-hoc test.

**Table 1 nutrients-13-02552-t001:** Blood GLP-1 concentration in WT and PASK-deficient mice.

	Blood GLP-1 Concentration (pM)	
Nutritional State	WT	*Pask ^−/−^*
NON-FASTED	4.13 ± 1.15	0.78 ± 0.38 *
FASTED (48H)	1.21± 0.82	4.29 ± 2.51

* *p*  ≤  0.05 WT vs. *Pask*^−/−^ by two-way ANOVA followed by Tukey post-hoc test.

## Data Availability

Not applicable.

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
