# Peer review of "Storage and Utilization of Glycogen by Mouse Liver during Adaptation to Nutritional Changes Are GLP-1 and PASK Dependent"

_nutrients, 2021, doi:10.3390/nu13082552_

Round 1
Reviewer 1 Report
Comments:
- Could the authors please justify the number of experiments for each experiment performed please i.e. how was their study powered?
- Could the authors please justify the use of parametric statistics in the description and analyses of the data please?
Author Response
Comments:
- Could the authors please justify the number of experiments for each experiment performed please i.e. how was their study powered?
We have revised the aspects of statistical treatments of results and some of the statistical tests have been modified accordingly. The details have been included in the section of material and methods section and also in each figure legend, depending on the case. In each case, the power of the statistical tests was calculated to be ≥ 0.8, according to the sample size used (n animals).
- Could the authors please justify the use of parametric statistics in the description and analyses of the data please?
As the reviewer suggest the statistical methodology was revised again and more details about the proceeding were included in materials and methods section, and also in the figure legends.
Parametric statistics are based on normal distribution, and we used it, since our data were tested and had a normal distribution and similar variances. First of all, we tested normality by Shapiro Wilk test, indicating that the data were normal. After that, we used the parametric test indicated in each figure (ANOVA and post hoc Tukey test or Fisher’s LSD test, t-test), since parametric tests are more powerful (we calculate the power of the used test and also was higher than 80%). Additionally, in the cases in which the calculated normality was slightly weaker, the significance was also verified by non-parametric tests (Mann-Whitney) and the significances were preserved, so we kept the test with more power (parametric).
The English has again been edited in the revised manuscript. We have used the services of a native English professional that we hope has enriched the writing.
Reviewer 2 Report
Over all the paper is well written with plenty of data supporting the results. However I have two comments. First, I have concern about how the Western Blot data were represented. In all the WB data, it is mentioned that n=3-5 was used per conditions. However most of the blots do not show this in the figure. In fact some of the blots show only one result per condition such as figure 2D WT non fasted +Ex 4 only one mouse and the result was very similar to WT non fasted -Ex4 # 3 mouse as seen from the figure. Therefore, for each of the WB data, I would like to see actual data with sufficient n numbers showing in the blot as well. My second comments is regarding glycogen metabolism related protein analysis. Effect on Gys2 was shown in the paper, but is the effect seen is specific for only Gys2 which is expressed in the liver or the effect could be seen in Gys1 as well? Was any such experiment done? If not, a prediction for the effect on Gys1 should be included in the discussion section.
Author Response
Over all the paper is well written with plenty of data supporting the results. However I have two comments. First, I have concern about how the Western Blot data were represented. In all the WB data, it is mentioned that n=3-5 was used per conditions. However most of the blots do not show this in the figure. In fact some of the blots show only one result per condition such as figure 2D WT non fasted +Ex 4 only one mouse and the result was very similar to WT non fasted -Ex4 # 3 mouse as seen from the figure. Therefore, for each of the WB data, I would like to see actual data with sufficient n numbers showing in the blot as well.
Perhaps it was not sufficiently explained. Our apologies. The data in the graph shows the means of the n indicated in the figure legend. The image shows only a representative gel of several carried out, since it is not feasible to make the total of samples from all the animals, with all the different treatments at the same time in a single gel. We can only include fifteen samples in each transfer (15-well gels is the maximum capacity) and we compare eight different conditions. This means that it is impossible to show 3 fasting WT samples, 3 fasting + ex WT, 3 fasting + ex WT, 3 fasting + ex WT and subsequent samples from the KO mice on a single gel. Therefore, we made at least 2-3 gels per protein to be tested, in order to obtain all the data and to be able to normalize and compare the results between different treatments and types of mice. We include the controls and the maximum samples by conditions in the same gel and make several gels with samples from different mice of each treatment and condition (WT and KO) adjusted to the capacity of the gel. Then, in each gel, the data was compared to non-fasting WT taken as 1. The result of joining the data from 2 or more gels is the mean that appears in the graphs. The bars show the average of 3-5 samples and a representative gel with several although it does not contain all the samples that were actually analyzed. A comment in relation to this circumstance has been included in materials and methods (“All the samples from different mice and treatment, were developed in 15-wells gels by SDS-PAGE”) and in the figure legends. Here, we are included some examples of gels used in the data processing to clarify the reviewer request (only visible for reviewer´s revision). Please see the attachment
My second comments is regarding glycogen metabolism related protein analysis. Effect on Gys2 was shown in the paper, but is the effect seen is specific for only Gys2 which is expressed in the liver or the effect could be seen in Gys1 as well? Was any such experiment done? If not, a prediction for the effect on Gys1 should be included in the discussion section.
We have not done any experiments focused on muscle specific GYS1 protein. However, according to the reviewer's suggestions, a couple of new comments have also been included in the discussion about GYS1:
“Our data also showed that the phosphorylation of liver GYS2, at Ser 640/641, and therefore the inactivation of this enzyme, was strongly blocked in PASK-deficient mice. It is in accordance with in vitro finding of the muscle glycogen synthase (GYS1) inactivation by direct PASK phosphorylation at Ser 641 [5], suggesting also a physiological role of PASK in muscle glycogen synthesis as we shown in the liver here”.
“Defective muscle glycogen synthesis has been reported in type 2 diabetes patients [60]. Phosphorylation of muscle glycogen synthase by PASK [5] or GSK3 is in addition the main regulatory modification for its catalytic activity [61]. Dephosphorylation of this site increased glycogen synthesis [62].”
The English has again been edited in the revised manuscript. We have used the services of a native English professional that we hope has enriched the writing.
Round 2
Reviewer 1 Report
The authors have satisfactorily responded to my comments.